# Study on Bainitic Transformation by Dilatometer and In Situ LSCM

**DOI:** 10.3390/ma12091534

**Published:** 2019-05-10

**Authors:** Xiaoyan Long, Fucheng Zhang, Zhinan Yang, Ming Zhang

**Affiliations:** 1State Key Laboratory of Metastable Materials Science and Technology, Yanshan University, Qinhuangdao 066004, China; longxiaoyanlxy@163.com (X.L.); zhm@ysu.edu.cn (M.Z.); 2Key Laboratory of Mechanical Reliability for Heavy Equipments and Large Structures of Hebei Province, Yanshan University, Qinhuangdao 066004, China; 3National Engineering Research Center for Equipment and Technology of Cold Strip Rolling, Yanshan University, Qinhuangdao 066004, China; zhinanyang@ysu.edu.cn; 4State Key Laboratory of Aviation Precision Bearings, Luoyang 471039, China

**Keywords:** bainite, in situ LSCM, phase transformation, kinetics, microstructure

## Abstract

This study investigates the bainitic transformation kinetics of carbide-free bainitic steel with Si + Al and carbide-bearing bainitic steel without Si + Al, as well as the phase transformation and microstructure through in situ high-temperature laser scanning confocal microscopy. Results show that bainitic ferrite plates preferentially nucleate at the grain boundary. New plates nucleate on previously formed ones, including two dimensions which appear on a plane where a three-dimensional space of bainitic ferrite forms. Nucleation on the formed bainitic ferrite is faster than that at the grain boundary in some grains. The bainitic ferrite growth at the austenite grain boundary is longer and has a faster transformation rate. The bainitic ferrite growth on the formed bainitic ferrite plate is shorter and has a slower transformation rate. The location and number of nucleation sites influence the thickness of the bainitic ferrite. The higher the number of plates preferentially nucleating at the original austenite grain boundary, the greater the thickness of the bainitic ferrite.

## 1. Introduction

Bainite possesses the most colorful structure in steel, and numerous forms of bainite have been discovered. The morphological characteristics of bainite are closely associated with the transformation temperature and alloying component. Carbide-bearing bainite is characterized by the precipitation of carbide within or between bainitic ferrite plates. Carbide-free bainite is obtained through the addition of alloying elements in an alloy composition that suppresses the precipitation of cementite during bainite transformation. The addition of Si and Al in steel inhibits cementite formation, and a large amount of carbon remains and enriches the retained austenite [1,2,3,4].

Characterization techniques and testing methods of materials have achieved rapid development, and some high-end technologies have been gradually applied to steel research. Bhadeshia and Caballero [5,6,7,8,9] explored the distribution of carbon and alloying elements in the bainitic ferrite phase and the retained austenite phase in bainitic steels through atom probe technology (APT), X-ray diffraction measurements, and transmission electron microscopy (TEM). They also investigated carbon supersaturation in bainitic ferrite in low-temperature bainite and found that supersaturated carbon is mainly trapped in Cottrell atmospheres and a bainite ferrite lattice. Pereloma et al. [10] studied the complex multiphase microstructures in transformation-induced plasticity steel through APT and found that the distribution of substitutional elements between a ferrite plate and austenite in carbide-containing bainite indicates a complete bainite transformation. Timokhina et al. [11] investigated Si-containing bainitic steel through neutron diffraction and APT, and observed carbon redistribution in high-density dislocations at the bainitic ferrite/austenite phase boundary. Many studies have explored bainitic transformation through in situ observations [12,13,14,15]. Bainitic transformation in low-carbon Si-containing steel was studied using in situ synchrotron X-rays. Results indicated that austenite was homogeneous prior to transformation, and carbon distribution became nonuniform as bainite plates formed [12]. Nucleation and morphological evolution in a Fe–C–Mn–Si superbainite steel was directly observed by high-temperature laser scanning confocal microscopy (LSCM). Bainitic ferrite nucleated at the grain boundary and in grains. Secondary bainitic ferrite sympathetically nucleated on the preformed bainite laths [13]. Sainis et al. [14] studied the nucleation and growth kinetics of bainitic ferrite plates and found that grain boundary nucleation was the dominant nucleation mode at all transformation temperatures. Additionally, the rate of nucleation also varied markedly among austenite grains [14]. Zhou et al. [15] studied the effects of martensite formation on nanostructured low-temperature bainite transformation and found that bainitic laths formed adjacent to a prior martensite plate, grew along the length direction, and hardly grew along the lateral direction. In summary, it is of great significance to study the phase transformation kinetics and microstructure by in situ observation.

This study aimed to design two types of steels, namely, carbide-free bainite with Si + Al and carbide-bearing bainite without Si + Al. The process of bainite transformation was observed by in situ LSCM. The kinetics and microstructure of bainite transformation were analyzed from the aspects of nucleation and the growth of bainite transformation.

## 2. Materials and Experimental Procedures

The chemical compositions of the tested steels used in this study are shown in Table 1.

**Testing of transformation kinetics**: Isothermal bainite transformation kinetic curves (TTT) and *M_s_* temperature were measured. The sample size and the test method were consistent with those in a previous study [16]. The size of the samples was machined to a cylinder with a diameter of 4 mm and a height of 10 mm. The samples were heated at a rate of 10 °C/s to reach the austenization temperature (930 °C) and held at this temperature for 10 min. The samples were cooled to air temperature at a rate of 30 °C/s, cooled to different temperatures, and held until bainitic transformation was completed. The TTT curve was plotted. 

**In situ characterization**: The samples for LSCM were machined to a cylinder with a diameter of 6 mm and a height of 4 mm. The top and bottom surfaces of the samples were polished conventionally to keep the measurement at a face level and minimize the effect of surface roughness. Investigations were conducted under a VL2000DX-SVF17SP laser scanning confocal microscope (LASERTEC, Yokohama, Japan). The samples were heated at a rate of 10 °C/ s to reach the austenization temperature (930 °C) and held at this temperature for 10 min. Afterward, the samples were cooled to 350 °C at a rate of 30 °C/s and held for 1 h. Bainitic transformation was recorded continuously during isothermal heat treatment. 

**Microstructure and phase characterization**: X-ray experiments were conducted using a D/max-2500/PC X-ray diffractometer (XRD, Rigaku, Tokyo, Japan) with unfiltered CuKa at 40 kV and 200 mA. The test was conducted in steps with a scan rate of 0.02° s^−1^ and stopped for 2 s per step. The volume fraction of the retained austenite (V_RA_) was calculated according to Equation (3) in the Ref. [17]. The microstructure was examined using a Hitachi 2010 TEM (Hitachi, Tokyo, Japan) operated at 200 kV.

## 3. Results and Analysis

### 3.1. Transformation Kinetics

Figure 1a depicts the TTT curves of the tested steels. *M*_s_ temperatures of the carbide-bearing bainitic steel and the carbide-free bainitic steel are 340 °C and 310 °C, respectively. The incubation period of the carbide-free bainitic steel is longer than that of the carbide-bearing bainitic steel. In a low temperature range (bainitic transformation), the addition of Si increases the activation energy of diffusion of carbon atoms, thus Si reduces the diffusion rate of carbon in austenite [18]. Then area of bainitic ferrite nucleation (carbon-poor area) is difficult to form, thereby extending the incubation period of bainite transformation. Si increases austenite strength and shear resistance at the nucleation of bainitic ferrite; therefore, a consequent downward shift of the curve is observed [19].

Carbide-free bainite requires a short transformation time, which is related to the driving force and amount of bainite during phase transformation. Figure 1b reveals the relationship between the free energy and temperature of the two steels as calculated by the MUGG83 software [20]. At any temperature, carbide-free bainite possesses a high free energy, which improves the diffusion rate and activation energy of carbon atoms for the phase transformation of austenite matrix atoms. Therefore, the time spent for carbide-free bainite transformation is shorter than that for carbide-bearing bainite transformation. According to the *T*_0_′ curve (as shown in Figure 1b, solid line), bainitic transformation occurs if the carbon content of residual austenite is lower than the *T*_0_′ curve. This was as also calculated by the MUGG83 software. *T*_0_ temperature is the intersection point of austenite and ferrite, which have the same chemical composition when the free energy is equal. The curve of all intersection points under different carbon contents is called the *T*_0_ curve. The *T*_0_′ and *T*_0_ curves are similar, but the effect of ferrite stored energy caused by a displacement transformation mechanism on the curve is considered based on the *T*_0_ curve. For carbide-bearing bainite, carbide precipitation consumes a larger amount of carbon and results in a lower carbon content in residual austenite as compared to carbide-free bainite. Therefore, the former bainitic transformation is complete and does not require further transformation time. Figure 1c illustrates the XRD curve of the tested steels. No austenite phase is observed in carbide-bearing bainite. The transformation content of carbide-free bainite reaches approximately 90% within a short time.

### 3.2. In Situ Observation

Figure 2 presents the process diagram of bainitic transformation. Grains 1, 2, 3, and 4 (G1, G2, G3, and G4, respectively) are the main observation objects. The time consumed for the formation of a new bainitic ferrite is defined as a reference time point of 0 s. The bainitic ferrite plates nucleate preferentially at the grain boundary of prior austenite (yellow arrows). As the transformation proceeds, new plates also nucleate on previously formed ones (red stars and red arrows, the former nucleates on the previously formed bainitic plate tip), including two dimensions which appear on a plane where a three-dimensional space of bainitic ferrite forms (blue circle). The blue circle in Figure 2b represents the latter case because of the lack of interface in the prior austenite grain. As the time is extended, the oblique longitudinal line disappears.

Figure 2f shows the relationship between the relative time and the number of nucleation sites in the grain boundary and bainitic ferrite. The number of nucleation sites markedly varies between G2 and G3. In particular, the number of these sites is larger in G2 than in G3. In the initial transition phase (60 s), bainitic ferrite randomly nucleates at the prior austenite grain boundary. After 80 s, the number of nucleation sites on the formed bainitic ferrite in G2 increases. This finding suggests that nucleation at the formed bainitic ferrite is faster than that at the grain boundary. In addition to the faster nucleation at the former bainitic ferrite lath, the number of possible nucleation sites increases. This may be due to more dislocation defects at the interface between bainite ferrite and austenite [21].

The reasons for nucleation faster at the formed bainitic ferrite are as follows. The analysis is related to autocatalytic nucleation, which is commonly associated with martensitic transformations [22]. The initial density of preexisting defects typically found in austenite is insufficiently large to explain the kinetics of martensitic transformation. The extra defects necessary to account for the transformation rates that are faster than the expected values are attributed to autocatalysis. Three mechanisms have been proposed for autocatalysis: stress-assisted nucleation, strain-induced autocatalysis, and interfacial autocatalysis. Initial nucleation is almost always confined to austenite grain surfaces, which presumably contain potent defects for nucleation. The initial formation of a plate of bainite may lead to appreciable plastic strains, and leads to an increase in the number and density of nucleation sites. In addition, nucleation on the formed bainitic ferrite aims to adapt to changes in shapes. Chu et al. found that the activation energy barrier for the nucleation of bainite ferrite at the phase boundary of austenite/martensite is only 0.000512 times that at the austenite grain boundary [23,24,25]. In our study, the phase boundary of the formed bainitic ferrite/austenite is similar to that of austenite/martensite, which exhibits a low activation energy barrier for nucleation.

Figure 3 shows the relationship between the growth rate of bainitic ferrite plates in the longitudinal direction at various nucleation sites and the relative time. The growth of bainitic ferrite plates formed at the original grain boundary is faster and the generated plates are longer as compared to previously formed plates. On the one hand, phase transformation occurs preferentially in the prior austenite grain boundary with sufficient space to allow the length direction to be unconstrained. The plates formed on the formed bainitic ferrite are formed after one or multiple divisions; thus, the space is constrained. On the other hand, carbon atoms in bainitic ferrite are diffused to the surrounding austenite during bainitic transformation. Bainitic ferrite plates that are preferentially formed in the austenite grain boundary can discharge carbon within a large area. The carbon solubility of the surrounding austenite in the bainitic ferrite formed at previously formed plates is higher than that of the original austenite, which leads to a high strength of surrounding austenite. Moreover, plastic relaxation is exhibited in the austenite adjacent to the bainitic ferrite during the former bainitic transformation. The dislocation debris generated in this process resists the advance of the bainite/austenite interface. All these factors restrict the growth of bainite ferrite plates in the longitudinal direction.

For nucleation at the original grain boundary, bainitic ferrite plates remain unchanged in terms of length after they rapidly elongate in the longitudinal direction. Their thickness continuously increases. For nucleation on previously formed plates, the plates elongate and the thickness increases simultaneously. After transformation occurs, the 3D diagram shows that the shape of the surface changes (even if the resolution is low), and this observation is affected by the plastic deformation of bainitic transformation (Figure 4). Plastic relaxation is, of course, ultimately responsible for the arrest in the growth of the bainite plates, as it is responsible for the sub-unit and sheaf hierarchies in the microstructure of bainite [22].

### 3.3. Relationship between Transformation, Kinetics and Microstructure

Figure 5 shows the relationship of the volume fraction of bainite, the transformation rate, and the transformation time of the two tested steels at 350 °C. The transformation rate of carbide-bearing bainite at the early stage is faster than that of carbide-free bainite due to the influence of the alloying element Si. The transformation rate of carbide-free bainite exceeds that of carbide-bearing bainite when the transformation time reaches 331 s. At 446 s, the amount of bainitic transformation in carbide-free bainitic steel exceeds that of carbide-bearing bainitic steel.

The transformation rate of carbide-free bainitic steel increases after the incubation period partly because the nucleation of the formed bainitic ferrite provides a favorable nucleation site. Figure 2 shows that the nucleation site of bainite transformation is not only at the original austenite grain boundary but also on the formed bainite ferrite. In G2, nucleation on the bainitic ferrite becomes the main model. After the relative time exceeds 80 s, nucleation on the formed bainitic ferrite occupies the main position. For carbide-free bainite, carbon diffuses to the retained austenite, forming a large concentration difference with the surrounding austenite, and this condition is conducive to secondary nucleation on the formed bainitic ferrite. In carbide-bearing bainitic transformation, in addition to carbon emission from bainitic ferrite, some carbides form, thereby producing less carbon in austenite. Therefore, the difference in the concentration of the formed carbon distribution is small.

Phase transformation is basically stagnant at the late stage (Figure 5b). The growth of bainite in the longitudinal direction is mostly independent of nucleation time and location. It terminates in the austenite grain boundary or the formed bainitic ferrite. A rapid migration interface with carbon spikes is formed in the length extension direction. However, the widening of the plates terminating within austenite grains results in a limited width of the bainitic plates. This finding implies that the transformation stasis can be caused by the stasis in the widening of the bainitic plates. Yang et al. predicted ferrite transformation with plate-like morphology, complete transformation in the length direction, and incomplete growth in the widening of plates. They proposed that global stasis in ferrite formation with a plate-like geometry is due to the local stasis in the widening of previously nucleated plates, not in lengthening; for isotropic ferrite morphology, stasis occurs in all growth directions [14,26]. 

Figure 6 shows the TEM images of the tested bainitic steels. Carbide-free bainite consists of bainitic ferrite and retained austenite (Figure 6a). The thickness of the bainitic ferrite plate in the carbide-free bainite is 133 ± 18 nm. Figure 6b displays the microstructure of the carbide-bearing lower bainite, which is composed of bainitic ferrite and carbides. The thickness of the bainitic ferrite plate in carbide-bearing bainite is 313 ± 34 nm. The perceived effect of temperature can be indirect because strength and nucleation rate are strongly dependent on temperature. A fine microstructure is related to strong austenite and high driving forces. In this study, the same temperature is observed in the two tested steels. Therefore, the effect of temperature is ruled out. The strength of supercooled austenite is measured by thermal compression (Figure 7). The results reveal that the strength of the supercooled austenite of carbide-free bainite is 14.2% higher than that of carbide-bearing bainite, and these observations are related to the alloying elements of steel [27]. Si and Al also increase the strength of austenite. The free energy of the former is higher than that of the latter. As such, the thickness of the bainitic ferrite plate is small.

Thickness is influenced by impingement between adjacent plates, and large nucleation rate and number correspond to a fine bainitic ferrite plate. Figure 5a shows that the transformation rate of carbide-bearing bainite without Si + Al is faster at the early stage than at other stages, and this observation is related to the short incubation period. After 330 s, the transformation rate of carbide-free bainite with Si + Al increases and it is 32.8% [(25.1%−18.9%)/18.9%)] higher than that in carbide-bearing bainite. This indicates that the number of nucleating plates on the formed bainitic ferrite is associated with the increases in phase transformation at this stage. In addition, nucleation sites are adjacent. Therefore, the growth of bainitic ferrite is restricted. As can be seen from Figure 6a, the thickness of bainitic ferrite plates at grain boundaries (red arrows) is smaller than that of the previously formed bainitic ferrite (yellow arrows). The two forms of bainitic ferrite are compared, and our results show that the greater the number of positions of preferential nucleation on the original austenite grain boundary in the early transformation stage, the greater the thickness of the bainitic ferrite. 

## 4. Conclusions

The transformation kinetics of carbide-free bainitic steel with Si + Al and carbide-bearing bainitic steel without Si + Al elements were studied. The process of bainite transformation was observed by in situ LSCM. The results are as follows:Carbide-free bainitic steel requires a long incubation period, because Si and Al hinder C diffusion, making the distribution of carbon atoms more homogeneous. Carbide-bearing bainitic steel exhibits a long transformation time.Bainitic ferrite plates nucleate preferentially on the grain boundary and new plates nucleate on previously formed ones. Compared to nucleation at grain boundaries, nucleation on previously formed bainitic ferrite is faster in some grains.The location and number of nucleation points have an important influence on the thickness of bainitic ferrite. The greater the number of positions of preferential nucleation on the original austenite grain boundary in the early transformation stage, the greater the thickness of the bainitic ferrite.

## Figures and Tables

**Figure 1 materials-12-01534-f001:**
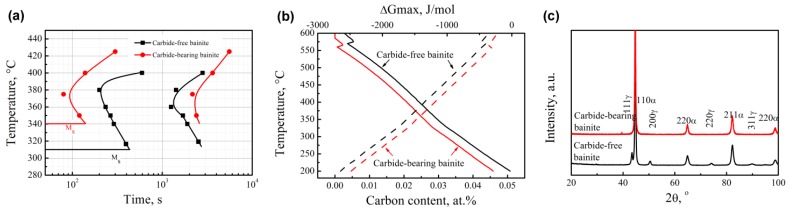
TTT curves (**a**), *T*_0_′ curves (solid lines) and relationships between the free energy and temperature (dotted lines) (**b**), and XRD curves (**c**) of the tested steels.

**Figure 2 materials-12-01534-f002:**
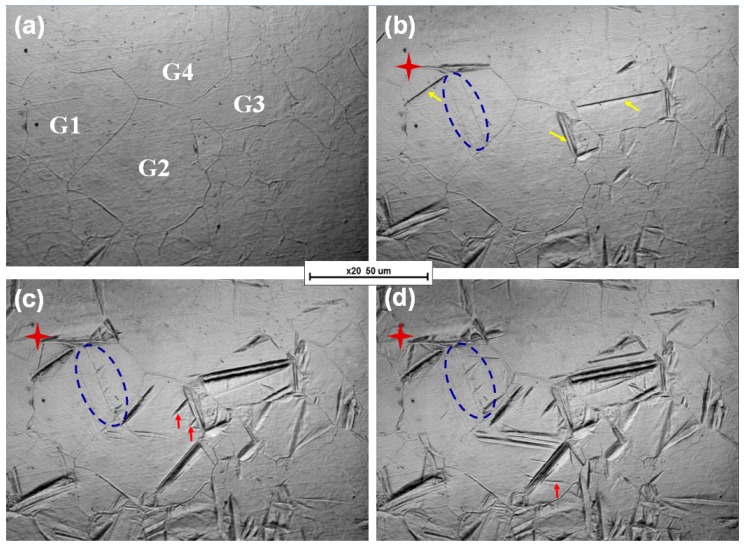
In situ laser scanning confocal microscopy (LSCM) of carbide-bearing bainite: (**a**) 0 s; (**b**) 60 s; (**c**) 110 s; (**d**) 130 s; (**e**) 180 s. (**f**) Relationship between relative time and number of nucleation sites. Note—GB: Grain boundary of original austenite; BF: Previously formed bainitic ferrite; yellow arrows: nucleation sites at GB; red stars and red arrows: nucleation sites on BF (the former represent nucleation sites on the previously formed BF tip); blue circle: nucleation site on BF that appears on a plane where a three-dimensional space of bainitic ferrite forms.

**Figure 3 materials-12-01534-f003:**
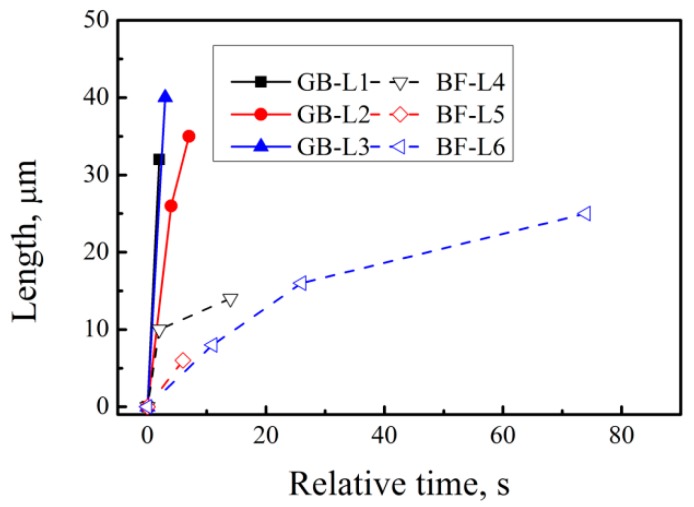
Grow rate of bainitic ferrite plates in the longitudinal direction at different nucleation locations.

**Figure 4 materials-12-01534-f004:**
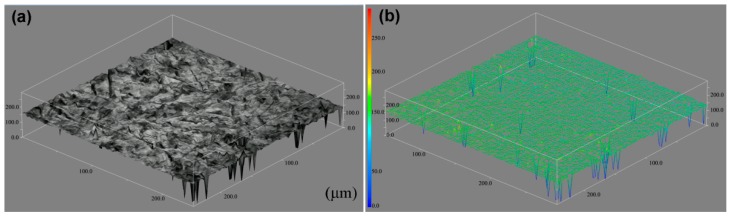
3D image of sample surface after bainite transformation.

**Figure 5 materials-12-01534-f005:**
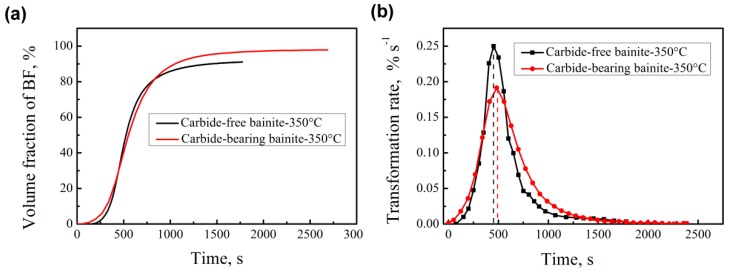
Relationship between the volume fraction of bainite (**a**), the transformation rate (**b**) and time.

**Figure 6 materials-12-01534-f006:**
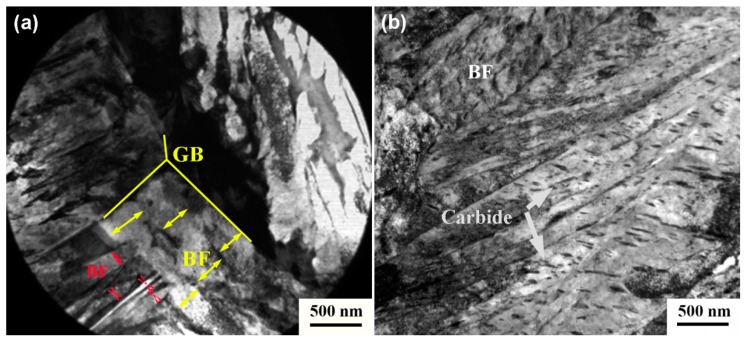
TEM micrographs of the tested steels: (**a**) carbide-free bainite; (**b**) carbide-bearing bainite.

**Figure 7 materials-12-01534-f007:**
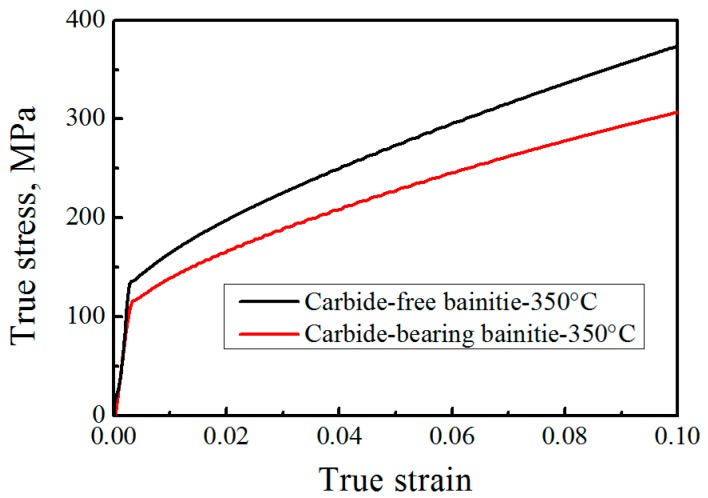
Supercooled austenite strength curve at 350 °C isothermal temperature.

**Table 1 materials-12-01534-t001:** Chemical composition of tested bainitic steels, wt.%.

Materials	C	Si	Mn	Cr	Ni	Mo	Al
Carbide-free bainite	0.34	1.48	1.52	1.15	0.93	0.40	0.71
Carbide-bearing bainite	0.34	0.01	1.61	1.24	0.96	0.45	0.04

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
