# Peer review of "Study on Bainitic Transformation by Dilatometer and In Situ LSCM"

_materials, 2019, doi:10.3390/ma12091534_

Reviewer 1 Report

Line 81: Please check if the heating rate 100°C/min is right. It differs from line 73. If there is no mistake please explain why both tests can be compared

Line 82: Please check cooling rate of 300°C/s. It is not in agreement with the cooling rate mentioned in line 75

Line 126+127: In addition to the faster nucleation at the former bainitc ferrite lath, their number and therefore possible nucleation sites increases. Please add a sentence to refer to that mechanism.

Line 144 (Fig2): Please explain the red stars, blue dashed circles and red and yellow arrows in the figure description.

Line 151: The sentence is unclear, I suppose you mean the following: ....at the original grain boundary is faster and the generates plates are longer than that...

Line 159: Please name the suggestate constraints on the growth of bainitic ferrite plates.

Line 185: I suppose it must be: ... In carbide-bearing bainitic transformation...

In general: Please explain the criteria to differ between nucleation on austenite grain boundary and former bainitic ferrite laths.

Author Response

Dear Reviewer:

Thanks for the kind review and the useful suggestions. We have revised the manuscript carefully. The point to point responses are described below and all corrections in the main text of the revised manuscript have been highlighted with yellow.

Point (1): Line 81: Please check if the heating rate 10°C/min is right. It differs from line 73. If there is no mistake please explain why both tests can be compared。

Response: Thank you very much for the comment. We have revised it. They are the same heat treatment process. The samples were heated at a rate of 10 °C/s to reach the austenization temperature (930 °C) and held at this temperature for 10 min.

Point (2): Line 82: Please check cooling rate of 300°C/s. It is not in agreement with the cooling rate mentioned in line 75.

Response: Thank you very much for the comment. We have revised it. They are the same heat treatment process. The samples were cooled to 350 °C at a rate of 30 °C/s and held for 1h.

Point (3): Line 126+127: In addition to the faster nucleation at the former bainitc ferrite lath, their number and therefore possible nucleation sites increases. Please add a sentence to refer to that mechanism.

Response: Thank you for your comments. We have revised it. In addition to the faster nucleation at the former bainitc ferrite lath, their number and therefore possible nucleation sites increases. This may be due to more dislocation defects at the interface between bainite ferrite and austenite [21].

[21] R. Rementeria, J.D. Poplawsky, M.M. Aranda, W. Guo, J.A. Jimenez, C. Garcia-Mateo, F.G. Caballero, Carbon concentration measurements by atom probe tomography in the ferritic phase of high-silicon steels. Acta Mater., 2017, 125: 359–368. https://doi.org/10.1016/j.actamat.2016.12.013.

Point (4): Line 144 (Fig2): Please explain the red stars, blue dashed circles and red and yellow arrows in the figure description.

Response: Thank you for your comments. We have revised it. Yellow arrows: nucleate at GB; Red stars and red arrows: nucleate on BF, the former nucleates on the previously-formed BF tip; Blue circle: nucleate on BF appear on a plane where a three-dimensional space of bainitic ferrite forms.

 Point (5): Line 151: The sentence is unclear, I suppose you mean the following: ....at the original grain boundary is faster and the generates plates are longer than that....

Response: Thank you for your comments. We have revised it. The growth of bainitic ferrite plates formed at the original grain boundary is faster and the generates plates are longer than that of previously formed plates.

Point (6): Line 159: Please name the suggestate constraints on the growth of bainitic ferrite plates.

Response: Thank you for your comments. We have revised it. On the one hand, phase transformation occurs preferentially in the prior austenite grain boundary with a sufficient space to cause the length direction unconstrained. The plates formed on the formed bainitic ferrite are formed after one or multiple divisions; thus, the space will be constrained. On the other hand, carbon atoms in bainitic ferrite are diffused to the surrounding austenite during bainitic transformation. Bainitic ferrite plates preferentially formed in the austenite grain boundary can discharge carbon within a large area. The carbon solubility of the surrounding austenite in the bainitic ferrite formed at previously formed plates is higher than that of the original austenite. which lead to a high strength of surrounding austenite. Moreover, it exhibits plastic relaxation in the austenite adjacent to the bainitic ferrite during the former bainitic transformation. The dislocation debris is generated in this process resists the advance of the bainite/austenite interface. All these factors restrict the growth of bainite ferrite plates in length.

Point (7): Line 185: I suppose it must be: ... In carbide-bearing bainitic transformation...

Response: Thank you for your comments. We have revised it.

Point (8): In general: Please explain the criteria to differ between nucleation on austenite grain boundary and former bainitic ferrite laths.

Response: Thank you for your comments. Bainite formation begins at austenite grain boundaries. At the initial stages of bainitic transformation, it leads to an increase in the number density of nucleation sites. Bainitic transformation continues autocatalytically at these newly created nucleation sites. In addition, Chu et al. found the activation energy barrier for the nucleation of bainite ferrite at the phase boundary of austenite/martensite is only 0.000512 times as much as that at the austenite grain boundary [23-25]. Details are as follows:

According to classical nucleation theory, the activation energy barrier for nucleation(G*) is lower when the nucleation occurs at α/g interface:

where ∆Gchem=GV(α)-GV(γ), GV is the Gibbs free energy per unit volume of ferrite (α), ∆Gstrain is the strain energy per unit volume of α and sα/g is the interfacial energy between α and austenite (γ). They found ∆Gchem and ∆Gstrain were the same for both nucleation at the interface of austenite/martensite and at grain boundary of austenite. The interface between ferrite and martensite was coherent, and the interfacial energy was determined to be 0.016 J/m2. The interface between austenite and ferrite is semi-coherent, and the interfacial energy between them was determined to be 0.20 J/m2. The activation energy barrier for the nucleation of bainite ferrite at the phase boundary of austenite/ martensite is only 0.000512 times as much as at an austenite grain boundary. Thus, phase boundary of the formed bainitic ferrite/ austenite  has low activation energy barrier for the nucleation.

Reviewer 2 Report

In this paper the bainite formation was observed in-situ under CSLM. From the alloys studied, readers expect discussion about the influence of Si and Al and carbide formation on bainite transformation. Unfortunately, the paper consists of sporadic observations and does not contain a sufficient amount of new facts or critical discussion.

Lines 94-95, ‘Si reduces the diffusion - - - ’

The authors are requested to show references.

The authors state that nucleation of bainite is faster at preformed bainitic ferrite, but growth is slower than those formed at austenite grain boundaries. The authors are requested to mention the possible mechanisms why this happens.

The authors discuss the thickness of bainite ferrite plates. They need to show the thickness of ferrite at grain boundaries and that on the preformed ferrite in a figure or table. Otherwise, readers do not recognize the facts being discussed.

Author Response

Dear Reviewer:

Thanks for the kind review and the useful suggestions. We have revised the manuscript carefully. The point to point responses are described below and all corrections in the main text of the revised manuscript have been highlighted with yellow.

Point (1): Lines 94-95, ‘Si reduces the diffusion - - - The authors are requested to show references.

Response: Thank you very much for the comment. We have added it.

[18] P. Zhao, F.Z. Xie, Z.G. Sun: Materials science essentials. 3rd edition, Harbin Institute of Technology Press, 2009.

Point (2): The authors state that nucleation of bainite is faster at preformed bainitic ferrite, but growth is slower than those formed at austenite grain boundaries. The authors are requested to mention the possible mechanisms why this happens.

Response: Thank you very much for the comment. We have revised it.

The reasons for nucleation faster at the formed bainitic ferrite are as follows. The analysis is related to autocatalytic nucleation, which is commonly associated with martensitic transformations [22]. The initial density of preexisting defects typically found in austenite is insufficiently large to explain the kinetics of martensitic transformation. The extra defects necessary to account for the transformation rates that are faster than the expected values are attributed to autocatalysis. Three mechanisms have been proposed for autocatalysis: stress-assisted nucleation, strain-induced autocatalysis, and interfacial autocatalysis. Initial nucleation is almost always confined to austenite grain surfaces, which presumably contain potent defects for nucleation. The initial formation of a plate of bainite may lead to appreciable plastic strains, and leads to an increase in the number density of nucleation sites. In addition, nucleation on the formed bainitic ferrite aims to adapt to changes in shapes. Chu et al. found the activation energy barrier for the nucleation of bainite ferrite at the phase boundary of austenite/martensite is only 0.000512 times as much as that at the austenite grain boundary [23-25]. In our study, the phase boundary of the formed bainitic ferrite/austenite is similar to that of austenite/martensite, which exhibits a low activation energy barrier for nucleation.

The reasons for slower growth than those formed at austenite grain boundaries are as follows. On the one hand, phase transformation occurs preferentially in the prior austenite grain boundary with a sufficient space to cause the length direction unconstrained. The plates formed on the formed bainitic ferrite are formed after one or multiple divisions; thus, the space will be constrained. On the other hand, carbon atoms in bainitic ferrite are diffused to the surrounding austenite during bainitic transformation. Bainitic ferrite plates preferentially formed in the austenite grain boundary can discharge carbon within a large area. The carbon solubility of the surrounding austenite in the bainitic ferrite formed at previously formed plates is higher than that of the original austenite, which lead to a high strength of surrounding austenite. Moreover, it exhibits plastic relaxation in the austenite adjacent to the bainitic ferrite during the former bainitic ferrite transformation.The dislocation debris is generated in this process resists the advance of the bainite/austenite interface, the resistance being greatest of strong austenite. The yield strength of the austenite must then feature in any assessment of plate size. All these factors restrict the growth of bainite ferrite plates in length.

Point (3): The authors discuss the thickness of bainite ferrite plates. They need to show the thickness of ferrite at grain boundaries and that on the preformed ferrite in a figure or table. Otherwise, readers do not recognize the facts being discussed.

Response: Thank you for your comments. We have added it. As shown in Fig. 6a,  the thickness of bainitic ferrite plates at grain boundaries (Red arrows) is finer than that of the previously-formed bainitic ferrite (Yellow arrows).

We tried our best to improve the manuscript and revised the manuscript in detail according to the comments. We appreciate for the Reviewer’s warm work earnestly, and hope that the correction will meet with approval.

I look forward to your positive response.

Thank you very much in deed.

Sincerely,

Zhang

Reviewer 3 Report

The work is of good quality and contributes to the understanding of the subject matter. The manuscript could be accepted for publication without any further revision.

Author Response

Thanks for the kind review and the useful suggestions. We have revised the manuscript carefully. The point to point responses are described below and all corrections in the main text of the revised manuscript have been highlighted with yellow.

Point (1): Ling 87: maybe it would be better to use: according to.

Response: Thank you very much for the comment. We have revised it. The volume fraction of the retained austenite (VRA) was calculated according to Equation (3) [17].

Point (2): Line 91: Fig. 1a.

Response: Thank you very much for the comment. We have revised it.

Point (3): Line 104-105: maybe it would be good to explain where is T0ʹ curve. maybe it would be good to explain what is T0'.

Response: Thank you for your comments. We have added it. According to the T0ʹ curve (as shown in Fig. 1b, solid line. It is calculated by the MUGG83 software. T0 temperature is the intersection point of austenite and ferrite with the same chemical composition when the free energy is equal. The curve of all intersection points under different carbon content is called T0 curve. T0ʹ and T0 curves are similar, but the effect of ferrite stored energy caused by displacement transformation mechanism on the curve is considered based on T0 curve.), bainitic transformation occurs if the carbon content of residual austenite is lower than T0ʹ curve.

Point (4): Line 170: maybe it would be good to make these figures visible, and write something more about them.

Response: Thank you very much for the comment. We have revised it. After transformation occurs, the 3D diagram shows that the shape of the surface changes (Even if the low resolution), and this observation is affected by the plastic deformation of bainitic transformation (Fig. 4). Plastic relaxation is of course, ultimately responsible for the arrest in the growth of the bainite plates, giving the sub-unit and sheaf hierarchies in the microstructure of bainite [22].

Figure 4. 3D image of sample surface after bainite transformation.

Point (5): Line 222: and it is

Response: Thank you very much for the comment. We have revised it. After 330 s, the transformation rate of carbide-free bainite with Si+Al increases and it is 32.8% [(25.1%-18.9%)/18.9%)] higher than that in carbide-bearing bainite.

We tried our best to improve the manuscript and revised the manuscript in detail according to the comments. We appreciate for the Reviewer’s warm work earnestly, and hope that the correction will meet with approval.

I look forward to your positive response.

Thank you very much in deed.

Sincerely,

Zhang

Round  2

Reviewer 1 Report

Accepted

Author Response

Dear Reviewer:

Thank you for your approve. We appreciate for your warm work earnestly. English language and style have been carefully checked.

Thank you very much in deed.

Sincerely,

Zhang

Reviewer 2 Report

Point (1)

Please show the page number.

There is no general consensus that Si or Al reduces carbon diffusion. Therefore, a refereed journal paper should be quoted.  Mn and Cr reduce carbon activity and hence delay carbon diffusion. Si and Al increase carbon activity. It could enhance carbon diffusion.

Point (3)

Concerning the thickness of bainitic ferrite plate, there is a strange statement in the Abstract.

‘The higher the number of plates preferentially nucleating at the original austenite grain boundary is, the coarser the thickness of the bainitic ferrite will be. The higher the number of plates nucleating on the formed bainitic ferrite is, the thicker the bainitic  ferrite plates will be.’

It does not say which is thicker. Micrographs in Fig. 2 seem to indicate that plates at austenite grain boundaries are thicker. What do the sentences in the Abstract mean?

Author Response

Dear Reviewer:

Thanks for the kind review and the useful suggestions. We have revised the manuscript carefully. The point to point responses are described below and all corrections in the main text of the revised manuscript have been highlighted with yellow.

Point (1): Please show the page number. There is no general consensus that Si or Al reduces carbon diffusion. Therefore, a refereed journal paper should be quoted. Mn and Cr reduce carbon activity and hence delay carbon diffusion. Si and Al increase carbon activity. It could enhance carbon diffusion.

Response: Thank you very much for the comment. We have revised it.

Si increase carbon activity. At higher temperatures, even within the temperature range of diffusive transformation (ferrite precipitation and pearlite transformation), the activity of Fe atom is large. The addition of Si improves the activity of carbon in austenite and promotes the diffusion of carbon in austenite. At lower temperatures, the activity of Fe and C atoms is very low. The addition of Si increases the activation energy of diffusion of carbon atoms, thus Si reduces the diffusion rate of carbon in austenite in temperature range of bainitic transformation [18].

[18] M.X. Zhang, J. Wang, M.K. Kang, The study on effect of silicon in steels (Part 1)-Influence of silicon on the continuous cooling transformation dynamics of undercooled austenite, Heat Treatment of Metals, 8 (1992): 3–7.

Point (2): Concerning the thickness of bainitic ferrite plate, there is a strange statement in the Abstract.‘The higher the number of plates preferentially nucleating at the original austenite grain boundary is, the coarser the thickness of the bainitic ferrite will be. The higher the number of plates nucleating on the formed bainitic ferrite is, the thicker the bainitic ferrite plates will be.’ It does not say which is thicker. Micrographs in Fig. 2 seem to indicate that plates at austenite grain boundaries are thicker. What do the sentences in the Abstract mean?

Response: Thank you very much for the comment. We have revised it. I agree with your opinion. The plates at austenite grain boundaries are thicker than that of nucleating on the formed bainitic ferrite.

We tried our best to improve the manuscript and revised the manuscript in detail according to the comments. We appreciate for the your warm work earnestly, and hope that the correction will meet with approval.

I look forward to your positive response.

Thank you very much in deed.

Sincerely,

Zhang